# Object-Centric Knowledge Representations in Hierarchical Planning for Decentralized Environments

## Abstract

This paper focusses on the under-studied problem of automated decentralized AI planning under severe communications and information-sharing restrictions between the planners during planning. Under such conditions, we will argue that traditional logic-based knowledge representations in planning systems are not only brittle and hard to manipulate for reasoning and deliberations but also not suitable for conveying important and maximal information with minimal communications between the planners or one's observations of another. We describe our proposed representation for object-centric knowledge graphs as a first-class element in our decentralized Hierarchical Task Network (HTN) based decentralized planning framework, called ARCADE. We present a number of preliminary experimental results, enabled by our new formalism.

## 1 Introduction

Traditional logic-based formalisms that are taken for granted for automated AI planning systems, in particular existing Hierarchical Task Network (HTN) planners, are not sufficiently powerful for knowledge representation and sharing when planning is decentralized under the conditions of limited and unreliable communications or observations in those planners. In particular, logical representation require a lot of knowledge engineering and may differ from one planner to another depending on the engineering process, even those different representations are aimed to convey the same semantics. Whether logical representations are propositional or a subset of first-order logic, a simple example of this issue is as follows. Suppose a planner uses a predicate form (on block1 table) and another planner uses (on-table block1). Even if the intended semantics captured between these two forms is the same, the planners that use one of these forms or the another cannot understand each other directly; they would either need additional knowledge engineering to align their representations or they will have to use automated translations from one form into another.

The issue is amplified when the automated planners are decentralized for collaborative problem-solving tasks and cannot communicate with each other fully and reliably. In that case, knowledge engineering for all automated planners

involved is further complicated by the fact that human experts may not know enough about the intricacies of each of the automated planners in the decentralized environment to timely address the knowledge representation issues between the planners.

In this paper, we argue that domain object-centric knowledge representations may have the potential to be better suited generally for AI planning, and in particular, automated planning for collaborative tasks in decentralized environments. We do not suggest that object-centric representation should replace logical ones; however, we argue that domain objects as first-class elements of an automated planner not only enable shared knowledge and situation awareness between decentralized planners but also may result in more powerful logical planning formalisms because they naturally carry semantics about attributes and relationships without explicitly representing such knowledge constructs.

Contributions of this work include:

- Development of a new planning formalism, based on knowledge graphs over first-class domain objects;

- Description of how we use this representation in our decentralized planning framework, called ARCADE (short for "Autonomy and Rationale Coordination Architecture for Decentralized Environments").

- Results of preliminary experiments to test object-centric representations in an abstracted version of military air operations domain.

In the subsequent sections, we first discuss the motivation of our work. We then discuss our SHOP2 HTN language requirements for plan repair with semantic soundness and completeness properties. Our preliminary experiments demonstrate the effectiveness of our approach.

Finally, we conclude by discussing our ongoing work and future research challenges for object-centric representations in HTNs and decentralized planning tasks.

## 2 Why are Objects Important?

Objects are the first-class entities in our physical surroundings. Logical representations and relationships, on the other hand, are knowledge artifacts humans create as a reasoning machinery for their surroundings. A PDDL-based or and HTN language representation does not exists in the world – if

a domain author changes the representation for planning with blocks, the behavior of an automated planner might change but the blocks remain in the environment. Similarly, if we remove the blocks from the environment, the logical artifacts would remain but would be semantically null.

Therefore, domain objects exist in the physical world. Primitive objectives are formed and transformed during their life time; more complex objects can be formed from primitives and may have capabilities that enable them to act and transform other objects. Either way, humans, other entities, and AI systems manipulate objects, not logical expressions in reality. A robot grasping a block has to reason about the block itself for its actuators in the primitive level not about some logical predicate. A human telling the robot to grasp the block must be able to point the block without translating into an artificial representation for the robot to understand what the human is referring to.

In human-to-human communications, objects carry intrinsic semantics that enable shared decentralized awareness without continually talking to each other. When mentioned or shown a familiar object, a human typically recognizes the properties and the capabilities of the object without those properties are articulated explicitly. Along with this implicit articulation often comes with the knowledge of to use the object or what to do with it. For example, if somebody is shown a hammer, that person would most probably know what this object is as well as its purpose. He or she will start to reason about all the situations and tasks it could be used in and how it can be useful for the task he or she might be performing at that moment.

Just as it does in human-to-human communications, having a shared intentional object-centric structure should, in and of itself, lessen the bandwidth required for exchanging information between the knowledge engineer and an AI planning system, as well as between a group of AI planners themselves. Objects could also entail expectations about their progress over time, including movement, resource expenditure, and expected goal completion times. Because all planners have an understanding of the object and its capabilities, information uptake is improved even without lengthy and reliable communication; updates and deviations can reference the shared object, rather than having to also include the full context as would be in traditional logical representations. This streamlines discussion and sets the focus to only what is critical: what has changed and what impact it will have on the overall plan.

## 3 ARCADE: Autonomy and Rationale Coordination Architecture for Decentralized Environments

In this section, we describe an example of object-centric representations as we are using them in our current work. As we have described in (Kuter, Goldman, and Hamell 2018) before, ARCADE is a decentralized planning architecture that allows multiple SHOP2 (Nau et al. 2003). In (Kuter, Goldman, and Hamell 2018), we have summarized ARCADE at a high-level but have not discussed the underpinnings of our object-centric representations in that paper. Below, we describe those with some replication from the previous paper for the sake of completeness.

Existing distributed and multi-agent planning systems (Torreño et al. 2017) typically focus on deterministic planning problems, with relatively simple models. They also typically assume a *single* overall planning task that must be distributed among multiple agents. Most practical applications for decentralized planning (e.g., military operations, UAV planning, and others) involve independent planners and reasoners that are responsible for accomplishing different tasks under large-scale uncertainty, while communicating their intentions and coordinating their actions. These planners often are not handed a single, large problem to be decomposed and then solved. Instead, these planners often receive their own planning problems to solve based on the organizational structures in which they are embedded (e.g., logistics and manufacturing systems separately plan to secure inputs and to make products). They may also receive additional tasks at runtime.

We designed our decentralized planning framework with the following objectives in mind:

1. *Asynchronous decentralization and planning:* Different SHOP2 instances must be able to receive their TSTN planning problems at different points in time during decentralized planning and they must be able to work on those problems concurrently, and each at its own pace.

2. *Task-centric assumption-based coordination:* SHOP2 instances must be able to exchange subtasks during planning, based on the assumptions each makes and whether or not a planner is capable of generating plans for specific tasks.

3. *Hierarchical localized plan adaptation and repair:* Each SHOP2 instance in the decentralized planning framework must use localized replanning and plan repair algorithms (Goldman and Kuter 2018) to provide consistency and correctness over its assumptions, which might be invalidated by the decisions and plans made by other SHOP2 instances.

ARCADE is summarized in (Kuter, Goldman, and Hamell 2018) so we will not repeat its motivations, formalism, and descriptions again in this paper. Instead, we will focus on the recent object-centric representations we have been working on for ARCADE. In particular, we formalize object-centric knowledge graphs as a first-class entity for the representation in the decentralized as follows. We assume that each planner in the framework has access to a finite set of domain objects, denoted as $\mathcal{B}$, in addition to the typical domain description it has. For example, PDDL-based domain descriptions typically required this information to be provided. In our HTN-based formalism, we added this requirement in the HTN domain descriptions in addition to the usual HTN methods, operators, and axioms as knowledge artifacts.

Each object in $\mathcal{B}$ is associated with a set of *capabilities*. Informally, a *capability* is defined as an attribute of the object that is useful for a task. More formally, a capability can be represented as a logical form (e.g., (robot r1) or (robot-holding r1 hand1 grasp)), could be implemented by using object-oriented data structures, or some other form desired.

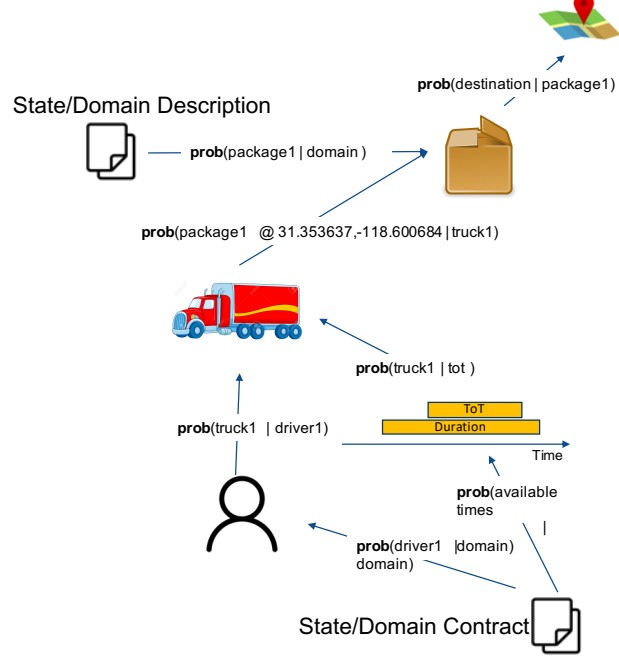

State/Domain Description

**prob**(package1 | domain )

**prob**(destination | package1)

**prob**(package1 @ 31.353637,-118.600684 | truck1)

**prob**(truck1 | tot )

ToT
Duration
Time

**prob**(truck1 | driver1)

**prob**(available times |

**prob**(driver1 |domain) domain)

State/Domain Contract

Figure 1: An example object-centric knowledge graph.

In ARCADE, we are currently using both logical and object-oriented representations for a capability.

ARCADE uses our SHOP2 HTN planning system for each decentralized planner. Traditionally, SHOP2 itself progress both the state of the planner that describe the logical facts to be true in the world and the task network being processed. During HTN planning, ARCADE also generates a knowledge graph over the objects that appear in the partial plan generated so far. Figure 1 shows an example of an object-centric knowledge graph in a delivery task. The nodes of the graph describe objects and the arrows between them describe dependencies. Note that some of the objects are primitive (e.g., truck, package) whereas some ore complex (e.g., time schedule of the truck). This information is already available in SHOP2's task network and state representations implicitly; however, in order to articulate it, the planner needs to extract the relevant information from its representations or needs to communicate the entire state or task representation with the other planners. Object-centric knowledge graphs enable the planner to do so seamless without any extra machinery.

ARCADE's object-centric knowledge graphs are probabilistic causal models. ARCADE leverages our previous work with probabilistic networks as reported in (Kuter et al. 2004; Kuter and Golbeck 2007; 2010) and uses a re-thinking of this work for object-centric knowledge graphs rather than traditional propositional causal graphs. We summarize this approach below.

As shown in Figure 1, nodes of the knowledge graph denote the objects involved in the representation. Some of these objects are primitive, i.e., they directly correspond to the input domain objects in $\mathcal{B}$. Others are complex as they are generated by the planner during the planning process, i.e.,

timelines or tasks. The edges (which are also called mechanisms) represent causal and inhibitory dependencies between objects. A mechanism $o1 \rightarrow o2$ between objects $o1$ and $o2$ is *causal* if the use of $o1$ increases $o2$'s probability of use in a plan, and it is inhibitory if the use of $o1$ reduces $o2$'s probability of use. Associated with each mechanism is a number between 0 and 1 to indicate the probability with which $o1$ causes or inhibits $o2$.

The causal and inhibitory mechanisms in the model are derived from the add and delete effects of the probabilistic action models ARCADE employs. Furthermore, each object in a causal model is associated with a special type of probability, called the *leak probability* for that object. Intuitively, an object's leak probability specifies the probability that the objects will appear in a plan even when none of its causes occurs in the world. In other words, a leak probability specifies the causes of an event that are not specified explicitly in the given causal model. Leak probabilities allow to work with incomplete causal models with unknown objects a priori and still be able to reason and compute the conditional probabilities over plans with the causal model. To calculate the probabilities of occurrence for the objects and dependencies of a causal model $M$, ARCADE implements the Noisy-OR rule (Pearl 1988) and its generalization Recursive Noisy-OR (RNOR) rule (Lemmer and Gossink 2004; Kuter et al. 2004 postponed).

By performing probabilistic belief updates over the probabilistic causal models of objects, ARCADE revises the probability that an object is to be used by a plan. This enables a SHOP2 instance in ARCADE to reason about the objects that might be used by the other SHOP2 instances in the framework and generate probabilistic dependencies in knowledge graphs that might include objects to be used by other planners. This naturally increases the communication bandwidth between planners even if the planners do not pass entire task networks or state representations around. Furthermore, it enables a planner to generate probabilistic plans even when it cannot talk to other planners. Doing so over states is a combinatorially complex endeavor; making assumptions over objects requires significantly simpler knowledge graphs and thus, enables efficient reasoning during planning.

## 4 Conclusions and Discussion

In this paper, we have argued for object-centric representations in automated planning, particularly for decentralized HTN planning. From a decentralized planning perspective, object-centric representations allow planners to develop shared knowledge awareness and reason about each other without communicating directly or when the communications are not reliable.

There are several challenges and research questions remain with developing object-centric representation for automated planning. One possible research direction is to learn the capabilities of objects from unstructured state information. Currently, we are assuming that the capability information is typically modeled in reference to the objects in the initial state and do not change. In reality, object capabilities could change over time or all of the capabilities of an object may not be given a priori.

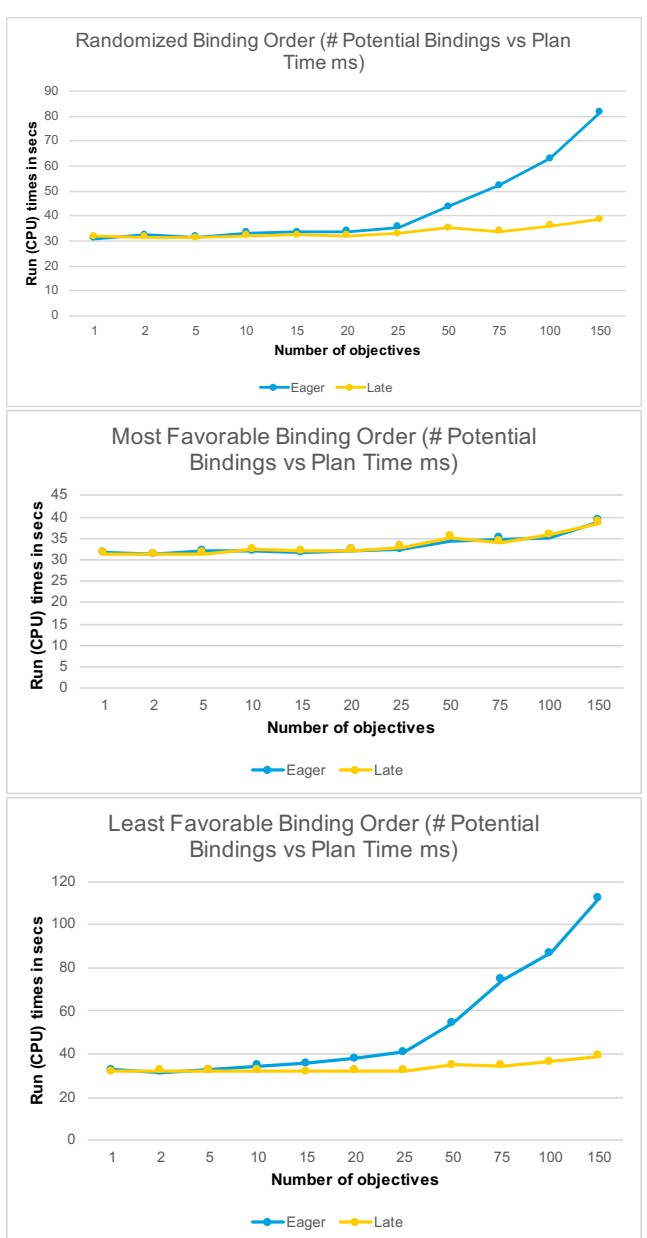

Figure 2: Preliminary experimental results with SHOP2 planning with late binding to objects. The first plot above is copied from (Kuter, Goldman, and Hamell 2018).

Another research direction, enabled by object-centric representations, is to explore *late-binding mechanisms* during planning. Almost all modern day classical planners are *eager binders*; i.e., the planning algorithm grounds variable symbols in its domain models as quickly and early as possible. Most state-of-the-art heuristic planners (e.g., FastDownward (Helmert 2006) and others) grounds variables during a preprocessing phase. Lifted planners such as SHOP2 binds variables on the fly but at the first opportunity during search.

Object-centric representations may enable a planner to use logical skolemization as in automated reasoning works (Genesereth and Nilsson 1987) in order to delay the binding while keeping the semantics and causal dependencies in a plan in tact. We have experimented with this idea in SHOP2, where SHOP2 delays binding a variable symbol for an object during planning and replaces it with a skolem function that specifies the properties, as constraints, of the constant that should be bound to that variable for a sound plan. After SHOP2 generates a plan with skolem functions in it as a "solution plan," ARCADE post-processes the plan and generates variable bindings according to the generated constraints during planning. Our preliminary experiments in a simplified logistics domain as shown in Figure 2 shows the potential benefits of this approach. In principle, however, post-processing may still fail to generate bindings successfully for some of the skolem functions. In that case, ARCADE treats the binding failure as a plan-failure discrepancy and may trigger its plan adaptation and repair process.

In addition to the all or none approaches above, it should be possible to develop heuristics for a planner to decide which object should be skolemized during planning. This will provide more informed manipulation of objects and reduce the post-processing or backtracking required at the end of a planning episode.

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
