# OpenReview forum: "Object-Centric Knowledge Representations in Hierarchical Planning for Decentralized Environments"
_icaps-conference.org/ICAPS/2019/Workshop/KEPS — KEPS 2019_

### Official Review · AnonReviewer2 · 2019-05-08
**Some good ideas about object-centric planning but additional clarifications needed across the paper**

**Rating:** 3
**Confidence:** 3

**Review:**

This paper proposes a new formalism for decentralised HTN planning based around the idea of object-centric knowledge graphs, where objects are treated as first-class citizens in the representation. Each object can be associated with a set of capabilities which connects attributes of an object to particular tasks. The formalism also enables causal and inhibitory relationships between objects to be modelled. A special type of probability, called the leak probability, specifies the probability that an object will appear in a plan even when none of its causes occur in the world, enabling the model to work with incomplete causal models with unknown objects. This formalism is described in the context of the ARCADE decentralised planning framework. Initial work on a late-binding mechanism is also mentioned, with preliminary results for the SHOP2 planner.

The idea of an object-centric representation for planning is certainly an interesting one, and one that offers certain advantages as the authors point out. However, I do have some concerns about the paper, especially with respect to clarifying certain aspects of the formalism that is being proposed. The writing in some sections of the paper could also be improved for clarity, and there are a number of relevant related works that could be added to the paper.

Comments and suggestions:

- In the Introduction, the authors seem to be motivated by two points: (1) traditional logic-based formalisms present certain problems, e.g., the need to solve the model alignment problem for distributed planning agents, and (2) object-centric representations offer a more natural model with certain inherent advantages. I think the authors have done a good job of justifying (2) but I'm less convinced about (1), which I don't see much discussion about in the paper. For instance, looking at the knowledge graph representation in Figure 1, which contains some fragments of logic-like representations, doesn't this raise some of the same problems discussed in the Introduction. E.g., how can we guarantee that even object labels are the same across a set of agents?

- I wasn't quite clear how domain designers actually specify object-centric knowledge graphs. There's a brief description near the end of page 2, and the description of Figure 1 is fairly good, however, some additional information about the actual formalism, its syntax, restrictions, etc. would be helpful for the reader. Given the criticism of traditional logic-based formalisms in the Introduction, this would also help clarify the approach.

- I didn't see any direct mention of the concept of affordance in the paper, although the authors certain mention ideas that are quire similar. In particular, the definition of "capabilities", which define attributes of objects that are useful for tasks, appears to be quite close to some representations of affordance.

- The contributions listed in the Introduction mention preliminary experimental results from an abstracted military air operations domain but this domain really isn't mentioned in the paper.

- It might be useful to mention the idea of late binding earlier in the paper, e.g., in the Introduction as one of the potential advantages of the approach. The first time we see it mentioned is right at the end of the paper, along with the experimental results. Maybe separate the Discussion from the Conclusions so it appears in the main body of the paper, rather than just tacked on at the end.

- In terms of related work, the authors may want to look at some of the work from the EU PACO-PLUS project (http://paco-plus.org/) which explored object-centric representations through the idea of Object-Action Complexes (OACs). See, e.g., the papers below which have some connections to planning:
Krueger et al. (2011), Object-Action Complexes: Grounded Abstractions of Sensory Motor Processes. RAS 59(10):740-757, doi:10.1016/j.robot.2011.05.009. Geib et al. (2006), Object Action Complexes as an Interface for Planning and Robot Control. Humanoids 2006 Workshop: Towards Cognitive Humanoid Robots.

Also, in terms of the connection between object-centric representations and human communication, the authors should look at the paper:
M. Steedman (2002). Plans, Affordances, and Combinatory Grammar. Linguistics and Philosophy, 25(5-6):723-753.

---

### Official Review · AnonReviewer1 · 2019-05-16
**The short paper illustrates some aspects of the Arcade planning system particularly focusing on the object-centric representation.**

**Rating:** 3
**Confidence:** 2

**Review:**

This paper is quickly written by expert author(s) in order to take part in the ICAPS workshop and consequently in the conference (I guess). I am sympathetic with the idea of going to ICAPS hence I would not complain too much on the strong dependence of this paper from a last year paper to another ICAPS workshop.

The point of the object-centered representation is a good one and I would really encourage the authors to "revise a bit"  the paper in its final version adding a couple of pages of formal materials to make it more self-contained.

I would also recommend them to situate the paper a bit more with the topic of the workshop.  Maybe they can mention in the intro as a minimum the work of GIPO and ItSIMPLE to create minimal connections instead of just having reference from the authors environment.

In general the paper is good for a "non official proceedings" workshop.